# Environmental Application of Biogenic Magnetite Nanoparticles to Remediate Chromium(III/VI)-Contaminated Water

**Yumi Kim and Yul Roh \*** 

Department of Earth and Environmental Sciences, Chonnam National University, Gwangju 61186, Korea; yumddalki@hotmail.com
* Correspondence: rohy@jnu.ac.kr; Tel.: +82-62-530-3458

**Abstract:** The physicochemical characteristics of biogenic minerals, such as high specific surface areas and high reactivity and the presence of a bacterial carrier matrix, make them promising for various applications. For instance, catalysts, adsorbents, oxidants, and reductants. The objective of this study is to examine the efficiency of biogenic magnetite nanoparticles (BMNs) that are produced by metal-reducing bacteria for removing chromium. Interactions between ionic chromium (Cr III/VI) and BMNs were examined under different pH values (ranging from pH 2 to pH 12) by using different doses of BMN (0–6 g/L). Chemically synthesized magnetite nanoparticles (CMNs) were used in the experiments for the purpose of comparing them to the BMNs. The results showed that the BMNs had higher Cr(VI) removal efficiency (100%) than the CMNs (82%) after a two-week reaction time. A lower pH and longer reaction time in the Cr-contaminated solution led to a higher Cr(VI) removal efficiency. The Cr(VI) removal efficiency by the BMNs in the Cr-contaminated groundwater was about 94% after a reaction time of two weeks. The BMNs that were coated with organic matter were more effective than the CMNs in leading to adsorption of Cr(III) with electrostatic interactions (82% versus 13%) and in preventing Fe(II) oxidation within the magnetite structure. These results indicate that the BMNs could be used to decontaminate ionic Cr in environmental remediation technologies.

**Keywords:** biomineralization; magnetite; chromium; remediation; groundwater

---

## 1. Introduction

Chromium (Cr) is primarily present in the environment in two oxidation states: Cr(III) and Cr(VI). While the trivalent Cr(III) is only toxic at high concentrations, the hexavalent Cr(VI) is a strong oxidizer that is toxic to humans and the environment at μg/L concentrations. Cr(III) exists primarily as a cation (e.g., $Cr^{3+}$, $Cr(OH)^{2+}$) in solution, while the Cr(VI) exists primarily as an anion (e.g., $HCrO_4^{-}$, $CrO_4^{2-}$, $Cr_2O_7^{2-}$), depending on the pH [1]. Iron-containing minerals are considered to be beneficial in chromium treatment in aqueous solutions because the minerals are predominantly available in nature. They play key roles in elemental recycling in the environment. Coupled Fe(II)–Fe(III) redox is considered to be highly attractive due to the enhanced electron-transfer mediator in the remediation process for heavy metals or radionuclides [2]. Consequently, Fe(II)-containing minerals, including pyrite, siderite, magnetite and mackinawite, have widely been investigated for Cr(VI) removal [3,4]. Among them, the magnetite has been evaluated and found to be a novel adsorbent for heavy metals. This is because combining nanoparticles adsorption and the magnetic separation technique is space-saving, cost-effective, simple-to-use and environmentally-sound compared to the present treatment technologies [5]. However, magnetite is not effective under basic conditions and Fe(II) in magnetite is highly susceptible to auto-oxidation [6]. Nano-sized magnetite is susceptible to

aggregation and can be toxic to the environment [7,8]. Recently, organic matter has been proposed to be coated on magnetite to reduce its toxicity and improve its stability [9,10]. Biological materials, such as plant tissues and by-products of edible mushroom, are used to coat organic matter with magnetite [8]. However, biogenic magnetite formed by microorganisms, does not require an organic coating process [11,12]. Biogenic minerals have good physicochemical characteristics, such as high specific surface area, high reactivity and the presence of bacterial carrier matrix, make them useful for various applications. For instance, they can be used as adsorbents, catalysts, oxidants and reductants [12,13]. A few research studies have reported that biogenic magnetite nanoparticles (BMNs) are successful in reducing Cr(VI) [14,15]. However, the removal efficiency of Cr(III) and Cr(VI) by BMNs under various experimental conditions has not been studied in details yet.

Therefore, the objective of this study was to determine the efficiency of BMNs produced by metal-reducing bacteria in Cr(III) and Cr(VI) removal from aqueous solutions under different pH values (ranging from pH 2 to pH 12) by using different doses of BMNs (0 to 6 g/L) at room temperature.

## 2. Materials and Methods

### 2.1. Preparation of Magnetite Nanoparticles and Characterization

Biogenic magnetite nanoparticles (BMNs) were synthesized using metal-reducing bacteria with 40 mM of akaganeite ($\beta$-FeOOH) as an electron acceptor and 10 mM glucose ($C_6H_{12}O_6$) as an electron donor in 1 L of growth media in accordance to the previous method [10]. The akaganeite ($\beta$-FeOOH) was prepared by using the following procedure: 10 M NaOH solution was slowly added into 0.4 M $FeCl_3 \cdot 6H_2O$ solution to precipitate the akaganeite ($\beta$-FeOOH) by gravity only with rapid stirring at pH 7. The suspension was aerated overnight by magnetic stirring for homogeneous oxidation and then washed three times with distilled water [16]. The metal-reducing bacteria (Geocha-1), which was a mixed culture, were enriched from the intertidal flat sediments of Suncheon in the Jeonnam province of South Korea. To obtain the anaerobic metal-reducing bacteria, the intertidal flat sediments were sampled from the shallow subsurface environments (5–10 cm deep) and stored in a 1 L glass bottle at a temperature of 25 °C. The metal-reducing microbes were enriched from the sediments by using Fe(III)-citrate and glucose under anaerobic conditions at room temperature (25 °C). For comparison, the chemically synthesized magnetites (CMNs) were prepared through the co-precipitation method [17] and used for experiments.

### 2.2. Chromium (III/VI) Removal

Cr(III)- and Cr(VI)-contaminated solutions were prepared by dissolving 50 mg/L of $CrCl_3 \cdot 6H_2O$ and $CrK_2O_4$ in distilled water, respectively. The Cr(III/VI) removal experiments were performed in 50 mL conical tubes. To examine the Cr removal by magnetite, 0.3 g of BMNs, and CMNs were added into the 50 mL of Cr(III)- or Cr(VI)-solution (pH 7) in a conical tube, respectively. All the samples in the conical tubes were shaken at 150 rpm to ensure complete mixing at room temperature. The reaction time ranged between 0.5 and 366 h. During the experiments, 1 mL of suspension was withdrawn with a 3 mL syringe at 30 min, 1 h, 24 h, 48 h, and 366 h and filtered through 0.2 $\mu$m filter to determine the rate of Cr(III/VI) removal by the BMNs or CMNs.

The effects of pH on Cr removal by the BMNs were evaluated through the addition of 0.1 g of BMNs into 50 mL of Cr(VI)-solution with different pH values (ranging from pH 2 to pH 12). To make the different pHs, the Cr(VI) solution was adjusted to pH 2, 4, 7, 10 and 12 with 0.5 M HCl or 0.5 M NaOH. No buffer was added in these experiments. For analysis, 1 mL of suspension was sampled at 0.5 and 48 h, respectively. In order to evaluate the in situ applicability, natural groundwater that had dissolved various inorganic nutrients was used as the Cr(VI)-contaminated groundwater (pH 7.8; initial Cr(VI) concentration at 50 mg/L) was spiked with $CrK_2O_4$. The groundwater collected from a military field range in Yonchon-gun, Gyeonggi-do, Korea was composed of the following cations and anions: $Mg^{2+}$ (8.34 mg/L), $Ca^{2+}$ (59.33 mg/L), $Na^+$ (17.58 mg/L), $K^+$ (2.69 mg/L), $Cl^-$ (39.05 mg/L), $NO_3^-$

(5.38 mg/L), $SO_4^{2-}$ (9.05 mg/L), $Al^{3+}$ (0.22 mg/L), $Cd^{2+}$ (0.003 mg/L), $Cu^{2+}$ (0.064 mg/L), and $Fe^{2/3+}$ (0.039 mg/L). The groundwater based test was performed using different doses (1–8 g/L) of BMNs in 50 mg/L Cr(VI)-contaminated solutions for 2 weeks.

## 2.3. Analytical Methods

Mineralogical characteristics of the magnetite precursor, akaganeite and transformed phases by microbial processes were examined by using X-ray diffraction (XRD) and transmission electron microscopy (TEM) with energy dispersive X-ray (EDS) analyses. The XRD was performed by using a X'Pert PRO (Panalytical, Almelo, Netherlands) equipped with Cu Kα radiation (40 kV, 20 mA) at a scan speed of 5 θ/min. The TEM analysis was conducted on a Phillips Tecnai F20 (Philips, Eindhoven, Netherlands) at an accelerating voltage of 200 kV to determine the morphology and elemental composition of the synthesized magnetite nanoparticle. The surface area of the freeze-dried BMNs was measured by using a Brunauer–Emmett–Teller (BET) surface area analyzer (ASAP-2020M, (Micromeritics Instrument Corp., Norcross, GA, USA). A zeta potential analysis via Dynamic light scattering (DLS) was performed by using a Zeta potential analyzer (Zetasizer Nano ZS, Malvern Instrument Ltd., Malvern, UK).

The concentrations of chromium were measured by using an inductively coupled plasma atomic emission spectroscopy (ICP-AES) on Optima 8300 (Perkin-Elmer, Waltham, MA, USA) in accordance with the standard method. Every experiment was run in triplicate and the average values were used in the graph development. The minimum detection limit for the ICP-AES of the metals was about 0.004 mg/L. The elemental composition and chemical oxidation state of the surface and near-surface species were investigated with an X-ray photoelectron spectroscopy (XPS) by using VG Multilab 2000 (VG Systems, East Grinstead, UK).

## 3. Results

### 3.1. Biogenic Magnetite Nanoparticles

The bacterial consortium (Geocha-1) reduced the akaganeite (β-FeOOH) to magnetite ($Fe_3O_4$) (Figure 1a,b). These BMNs were spherical in shape with sizes of around 5–15 nm in diameter (Figure 1d). The BMNs synthesized by bacteria were coated with organic materials that had a functional group, such as carboxyl (-COO⁻) in the previous study [11]. They were regarded as extracellular polymeric substances (EPSs) that resulted from microbial metabolism [11]. The surface area of the freeze-dried BMNs was measured at 101 $m^2/g$, which was similar to that (103 $m^2/g$) of biogenic magnetite nanoparticles formed by the *Geobacter* species [18]. The point of zero charge, $pH_{pzc}$, of these BMNs was around 6.8 (Figure 1e).

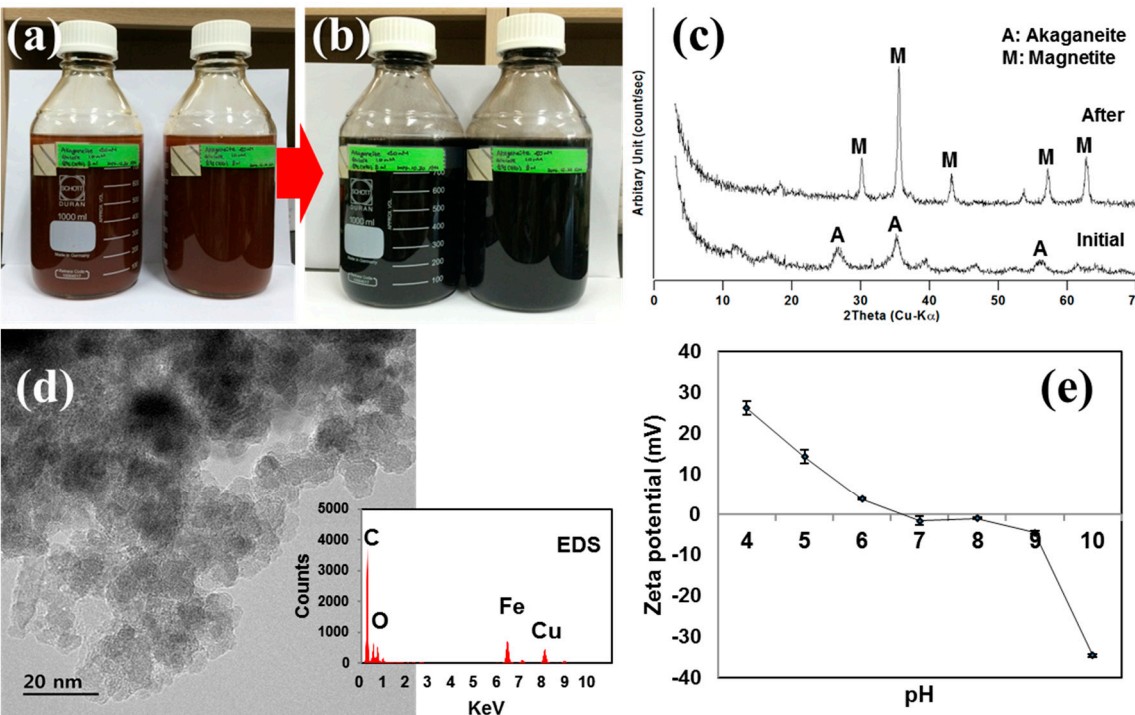

**Figure 1.** The colour changes due to phase transition of (**a**) brown akaganeite to (**b**) black magnetite, (**c**) XRD patterns showing mineral transformation, (**d**) TEM-EDS of biogenic magnetite nanoparticles, and (**e**) Zeta potential of BMNs.

### 3.2. Cr(III) and Cr(VI) Removal

Fifty milliliters of 50 mg/L of Cr(III) and Cr(VI) contaminated water were reacted with 0.3 g BMNs or CMNs at a pH of 7. The effect of time on the removal efficiency of the Cr(III/VI) by magnetite nanoparticles was as shown in Figure 2. After injecting magnetite into the Cr(III) contaminated water, Fe was rapidly released to 207.7 mg/L from the CMNs, but the iron was leached out from the BMNs at less than 3 mg/L within 30 min (Figure 2a). The Cr(III) concentration showed the most rapid changes within 48 h. Approximately 73% of the Cr(III) in the aqueous solution was removed by the BMNs, but it was about 11% by CMNs. After two weeks of the reaction time, the final removal rate of the BMNs increased to 82%, but it was slightly changed to 13% in the CMNs. Therefore, the high removal efficiency of Cr(III) by the BMNs was considered to be favorable for the $Cr^{3+}$ adsorption because the organic matter coated on the surface of the BMNs was negatively charged. In addition, the amount of iron leached from the magnetite had no effect on the removal of Cr(III).

On the other hand, Cr(VI) was removed by both the CMNs and BMNs (Figure 2b). As in the case of Cr (III), the Fe of magnetite in the Cr(VI) contaminated water was rapidly released to 174.7 mg/L from CMNs within 30 min, but it was released in little amounts from the BMNs (Figure 2b). At the same time, over 50% of the Cr(VI) was removed by CMNs during the first 30 min of reaction time and the additional 20% was removed after the 30 min of reaction time. The correlation between the release of Fe(II) and removal of Cr(VI) was observed when using the CMNs. Baig et al. (2014) have revealed that a high capacity of Cr(VI) adsorption and redox in aqueous solutions is dependent on the release of Fe(II) from magnetite, which is oxidized into Fe(III), resulting in reduction of Cr(VI) to Cr(III) [2]. However, the Cr(VI) removal efficiency by the BMNs was gradually increased as the reaction time increased. Lower Cr(VI) removal efficiency by the BMNs than that by CMNs was carried on until at 48 h reaction. After a reaction time of two weeks, the BMNs finally showed a higher Cr(VI) removal efficiency (100%) than the CMNs (approx. 82%). Thus, about 40% of the Cr(VI) was additionally removed by BMNs after 48 h of reaction, while only a small portion of the additional removal (approx. 5%) was achieved by the CMNs after 24 h of reaction time. The Cr(III/VI) concentrations were not

measured between 48 h and two weeks, but the data's absence did not have a significant effect on understanding the concentration decrease tendency since the major reaction of Cr removal occurred within 48 h. Such slower removal rates of the Cr(VI) by the BMNs compared to that by CMNs might be attributed to organic matter on BMNs. It might have prevented oxidation of the Fe(II) within the magnetite structure in the aqueous solutions. In effect, lower amounts of iron leaching (2 mg/L) from the BMNs compared to that (112 mg/L) from the CMNs was observed after 48 h of reaction in Cr(VI)-solution based on ICP-AES analysis (wavelength of 238.2 nm) (Figure 2). Therefore, it was considered that the adsorption and/or redox of the Cr(VI) on the surface of the BMNs coated with organic matter were more important than the effect of the leached Fe. The oxidation of the Fe(II) in magnetite can weaken the magnetism so that the structural stability due to the low Fe leaching of BMNs will be effective when the magnetite is recovered by magnetic separation after reacting with contaminants. After 48 h of reaction, the nanoparticles recovered showed XRD peaks without mineralogical changes during the contact time with Cr(VI) (Figure 3a). Moreover, the TEM and STEM elemental spot map images of the nanoparticles showed Cr adsorption on the surface of the particles (Figure 3b–e). The results of the EDS revealed that 1.2 wt% of Cr adsorbed onto the surface of BMNs after 48 h of reaction (Figure 3b,e).

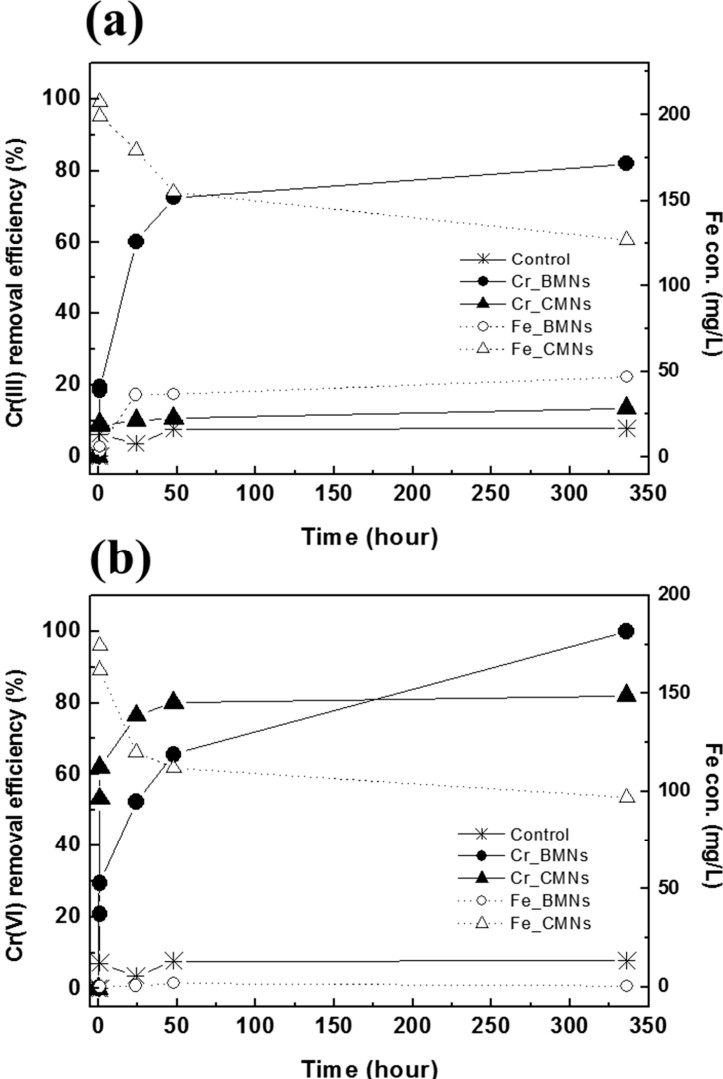

**Figure 2.** Cr removal efficiencies and Fe concentrations during two weeks in (**a**) Cr(III) and (**b**) Cr(VI) by BMNs and CMNs.

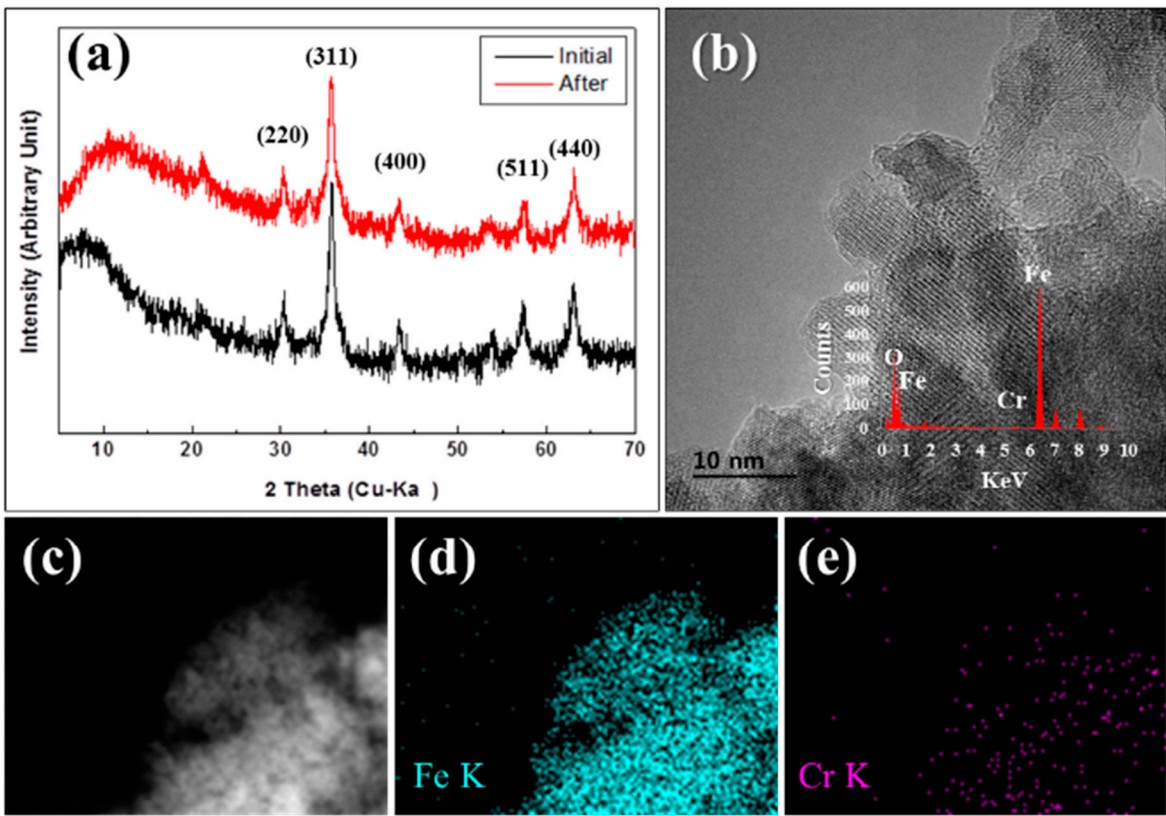

**Figure 3.** (**a**) XRD patterns, (**b**) TEM-EDS images, (**c**) backscattered electron image, (**d**) STEM-EDS elemental spot map of Fe, and (**e**) STEM-EDS elemental spot map of Cr for BMNs reacted Cr(VI)-contaminated water for 48 h.

To further understand the interaction between the Cr(VI) and BMNs, after reaction with Cr(VI) for 48 h, the sample was analyzed by XPS. An XPS spectra of C 1s, Fe 2p core level gave further proof on the chemical structure of the BMNs coated with organic matter (Figure 4). Before reaction with Cr, the Fe $2p_{3/2}$ peaks appeared at 710.2 and 710.5 eV in the BMNs and CMNs, respectively. This indicated that Fe was present as $Fe_3O_4$ [19]. The BE values at ~710 eV were reported for Fe(II)/Fe(III) oxides. The values of ~711 to ~712 eV were more consistent with the values for Fe(III) hydroxide phases [20].

In particular, the C 1s peak at 288.5 eV belonging to carboxylate (-COO$^-$) moiety [21] was detected in the BMNs but not in the CMNs (data not shown). The XPS spectra of the chromium from the adsorbent surface after reacting with Cr(III/VI)-solutions are shown in Figure 4. Fe $2p_{3/2}$ peaks slightly shifted at 710.6 and 710.9 eV in the BMNs and CMNs, respectively. The increase of the binding energy of the Fe $2p_{1/2}$ had been previously suggested as an oxidation of Fe(II) to Fe(III) or substitution of Cr(III) for Fe(III) in the Cr-substituted magnetite [5,22]. However, further experimental evidences are needed. The Cr 2p photoelectron peak indicates the existence of at least two Cr chemical species, which are Cr(VI) and Cr(III). In previous studies, XPS spectra have revealed peaks at ~577 eV for Cr(III) and at ~579 eV for Cr(VI) [23]. The BE values of the Cr $2p_{3/2}$ peak on the BMNs reacted with the Cr(III/VI)-solutions, while on the CMNs, they reacted with Cr(VI) fall within 576.6 to 577.4 eV. This can be attributed to the Cr(III), which corresponds to species that participates in $Cr_2O_3$, CrO(OH) or $Cr(OH)_3$. In particular, the CMNs reacted with the Cr(VI) and showed a peak at 579.5 eV corresponding to the Cr(VI) as $CrO_4^{2-}$ ion or in $CrO_3$ [24]. This indicated that Cr was adsorbed onto the CMNs as Cr(III) and Cr(VI), suggesting that some adsorbed Cr(VI) anions were reduced to Cr(III) on the surface of the CMNs. Adsorbed Cr(VI) on the BMNs were dominantly observed as Cr(III) reduced phases.

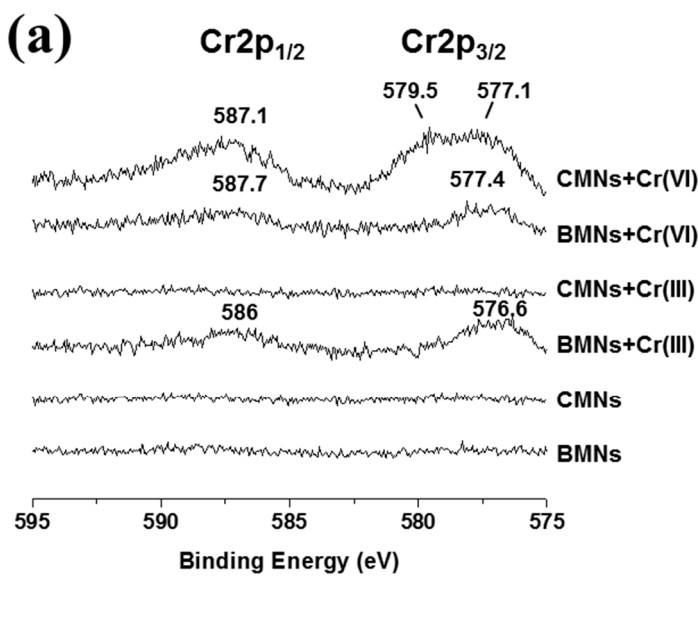

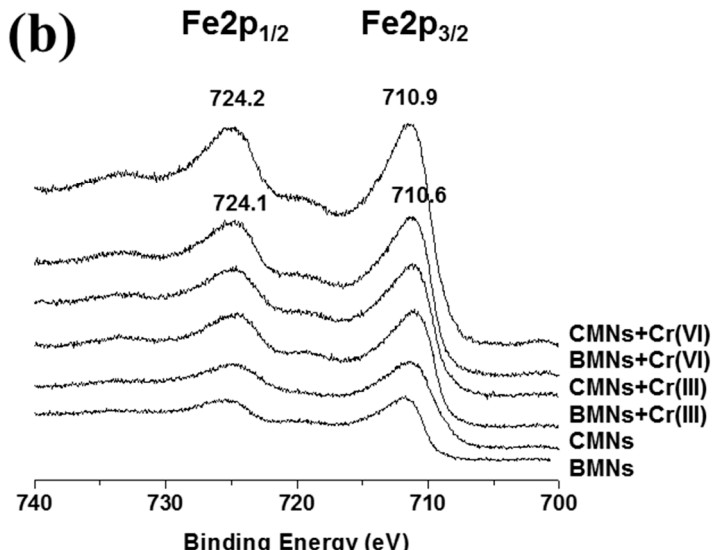

**Figure 4.** (**a**) Cr 2p and (**b**) Fe 2p XPS spectra of CMNs and BMNs reacted Cr(III/VI).

In comparison to the BE values reported in previous studies, a lower BE value of 576.6 eV for the BMNs reacted with the Cr(III)-solution in this study and is more comparable to the Cr(III) oxides. On the other hand, the slightly higher values of other samples with the BE values of 577.1–577.4 eV were more consistent with the Cr(III) hydroxides [20]. However, the CMNs that reacted with Cr(III) showed no Cr peak, which indicated that there was no adsorption during the process. Consequently, these results showed that both magnetites had a high capacity of Cr(VI) adsorption and redox in the aqueous solution. However, the CMNs had rapid reactions by Fe(II) oxidation while the BMNs showed high efficiency in the long run, indicating that the main mechanisms involved in adsorption and redox were different. These results indicated that the Cr(VI) in the solution might have been adsorbed onto the surface of the CMNs and then reduced to Cr(III) via the redox process [25]. Moreover, a large amount of iron that was released would have accelerated the reduction of the Cr. However, the BMNs is a little bit more complicated because it is related to organic complex, which might have originated from extracellular polymeric substances (EPS). The BMNs would have provided two reactive sites, which are the mineral surface and organic layer. The surface of the organic coating had a repulsive

force with the Cr (VI) ions, so the adsorption was disadvantageous [9]. Uncoated mineral surfaces might have been used for Fe release and Cr adsorption and redox.

### 3.3. Effect of pH for Cr(VI) Removal

As the results of experiments using different pH solutions, the lower the pH of contaminated solution, the higher the Cr(VI) removal efficiency. The uptake of Cr(VI) by the BMNs for 48 h was highly dependent on the pH (Figure 5). The graph in Figure 5 shows the average of the measured pH values after one hour and 48 h, and the range bars show the Cr removal rate after one hour (minimum value) and 48 h (maximum value) reactions, respectively. The small differences in the range bars at pH 2 and 12 indicate that Cr removal occurred in a stable and rapid manner in acidic (pH 2) and alkaline (pH 12) environments. However, in the range of pH 4 to 10, the Cr removal rate tended to increase with the reaction time. The maximum removal efficiency using the BMNs was about 77.5% for the initial Cr(VI) concentrations of 50 mg/L at pH 2. The amount of Cr(VI) removed from the solution decreased with increasing pH. The removal efficiency was less than 50% at pH > 7. When the pH was less than $pH_{pzc} = 6.8$, the surface of the BMNs was protonated and the positively charged surface attracted Cr(VI) ions. The variation in removal efficiency at different pH values may be attributed to affinities of the BMNs for different species of Cr(VI) existing at acidic pH values, namely the $H_2CrO_4$, $HCrO_4^-$, $CrO_4^{2-}$, and $Cr_2O_7^{2-}$ [26]. Optimum adsorption occurred where the pH < 4 (Figure 5a). With increase in pH, the amount of Cr(VI) removal decreased because of higher concentrations of $OH^-$ ions present in the reaction mixture, which might have competed with the Cr(VI) species for the adsorption site [5]. At pH > $pH_{pzc} = 6.8$, the BMNs surface was negatively charged, thus increasing electrostatic repulsion with the negatively charged Cr(VI) species. However, there was neither complete removal (100%) of Cr(VI) at strong acidic pH value (pH 2) nor complete remain (100%) of Cr(VI) at a high pH (pH 10). Thus, these imply that some other factors might have played a role. In addition, the presence of organic matters on the BMNs was probably preventing Fe(II) oxidation from the magnetite during the short time and leading to continuous reactions to achieve high efficiency of Cr(VI) removal during the long-term period (Figure 5b). Ion exchange was also expected to occur under the alkaline conditions. According to Dimitri et al. (2000), when the affinity of $CrO_4^{2-}$ with minerals is higher than that of $OH^-$, the $CrO_4^{2-}$ can exchange with $OH^-$ from the surface of the hydrolyzed mineral at a high pH [27]. Therefore, the Cr(VI) removal by the BMNs might be affected by the functional organic matter coated on particles, redox reaction, surface charge, and ion affinity depending on the pH.

### 3.4. Effect of Common Ions on Cr(VI) Removal

In order to evaluate the in situ applicability, various inorganic nutrients dissolved in natural groundwater were used as the Cr(VI)-contaminated groundwater. The presence of common ions coexisting with the Cr(VI) in groundwater might have competed for available adsorption sites. Although not all solutes will compete for the exact adsorption sites, the presence of other solutes might reduce the adsorption of any given solute to some degree [5]. In the Cr(VI)-contaminated groundwater, the main cations were $Mg^{2+}$, $Ca^{2+}$, $Na^+$, $K^+$, $Al^{3+}$, $Cu^{2+,}$ and $Fe^{2/3+}$ while the main anions were $Cl^-$, $NO_3^-$ and $SO_4^{2-}$. As a result, the Cr(VI) removal efficiency by the BMNs in groundwater was about 94% after two weeks of reaction (Figure 5b). The Cr (VI) reduction rate increased when a dose of magnetite was added. When the amount of magnetite added was the same at 6 g/L as in distilled water, the removal rate of the Cr in distilled water was 100%, while that of groundwater was decreased by 20%. When the amount of magnetite was increased to 8 g/L, the Cr removal rate was 94% in contaminated groundwater water. Therefore, in order to obtain a high Cr removal efficiency as in the distilled water, 30% or more of magnetite should be added in contaminated groundwater water. However, such small competitive influences of these ions could be negligible for in situ remediation [5]. As shown in Figure 6, the longer the reaction time, the higher the Cr(VI) removal efficiency.

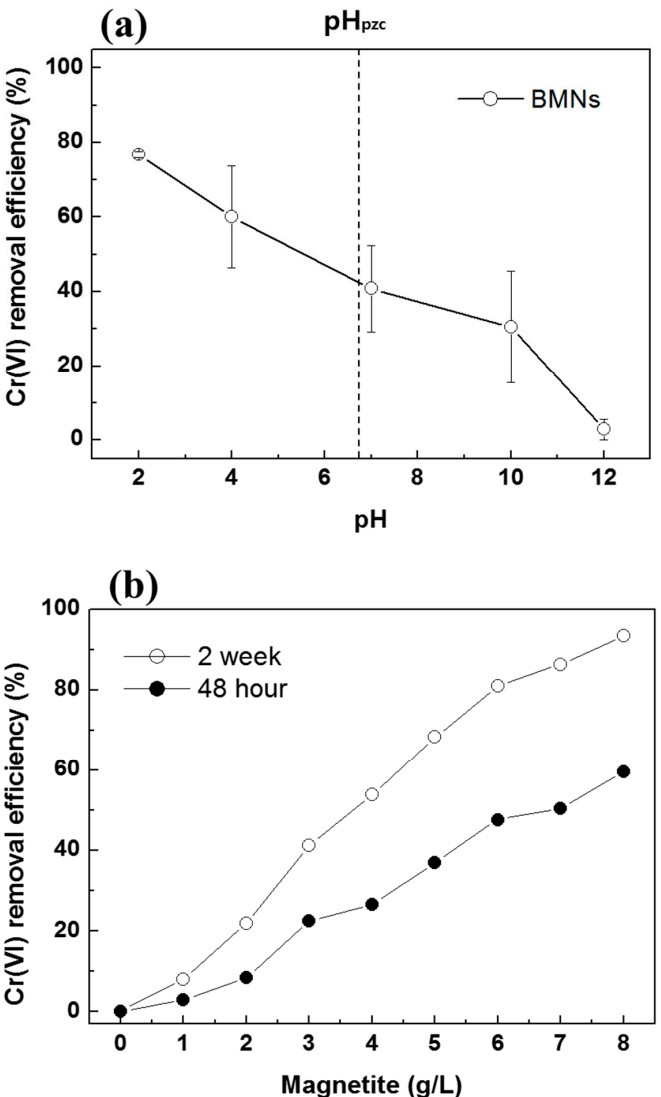

**Figure 5.** Cr(VI) removal efficiency by BMNs at (**a**) different pH values of solutions or (**b**) different reaction time in Cr(VI)-contaminated groundwater.

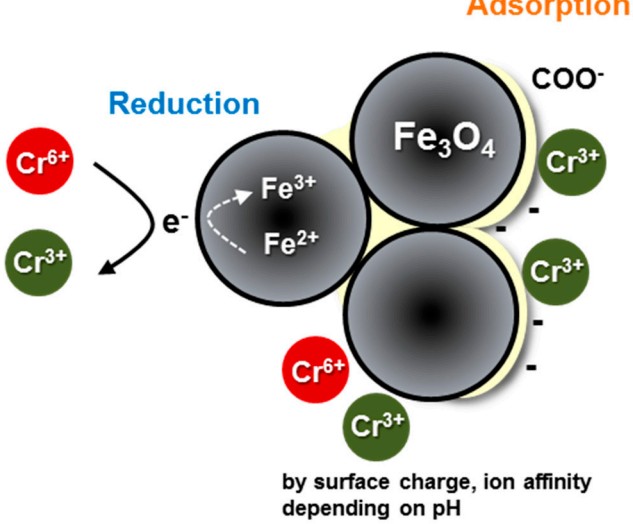

**Figure 6.** Schematic illustration of the mechanism of Cr(III/VI) removal by biogenic magnetite nanoparticles.

## 4. Conclusions

Despite the high Cr(VI) removal efficiency in acidic contaminated solutions, the BMNs were relatively stable at a pH range of 2–10. Therefore, they might be used in most natural and wastewaters. The BMNs that were coated with organic matters were effective in leading the adsorption of Cr(III) with electrostatic interaction, and to prevent fast oxidation of Fe(II) within the magnetite structure enhancing Cr(VI) reduction efficiency in the long-run. These results indicate that the combined effect of the adsorption and redox on the BMNs for Cr(VI) removal are accomplished by two reaction sites of mineral surface and functional organic layers (Figure 6). In addition, the Cr(VI) removal using BMNs may be affected by surface charge and ion affinity depending on the pH. Therefore, BMNs, as functional magnetite-organic complex nanoparticles, may be effective for long-term in-situ remediation for the Cr-contaminated sites with various pH ranges by minimizing the loss of their magnetic recoverability. They also have high potential to applicability for riding heavy metals in environmental remediation technologies.

**Author Contributions:** Y.K. designed the experiments, performed analysis on all samples, interpreted the data and wrote the manuscript; Y.R. conceived the original idea, provided critical feedback and acted as the corresponding author.

**Funding:** This research was supported by the Basic Science Research Program of the National Research Foundation of Korea and funded by the Ministry of Education (NRF-2016R1D1A1A09917588).

**Acknowledgments:** We are grateful to Moon and Bae at KBSI-Gwangju Branch for SEM/TEM-EDS and ICP-AES analyses, and Jung at CCRF in Chonnam National University for XRD and XPS analyses.

**Conflicts of Interest:** The authors declare no conflict of interest.

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
