# Peer review of "Environmental Application of Biogenic Magnetite Nanoparticles to Remediate Chromium(III/VI)-Contaminated Water"

_minerals, doi:10.3390/min9050260_

Round 1

Reviewer 1 Report

attached

Author Response

Thank you for your comments.

Reviewer 2 Report

Minerals-467990

In this work, the authors synthesized BMNs for Cr(VI) reduction and removal. The obtained results showed the higher reactivity of BMNs over CMNs, and the authors suggested that BMNs could be used to decontaminate ionic chromium in environmental remediation. However, due to the following reasons, I suggest a major revision is necessary before it can be accepted for publication.   

1.   Although the BMNs have better reactivity, the CMNs have the advantage of simple synthesis process and scalable production. Moreover, the authors did not give detailed interpretation why the CMNs have better efficiency. Is it just because of the particle size? Moreover, natural magnetite is much cheaper and can be applied in Cr removal. In this case, I can hardly see the advantage of BMNs over CMNs and natural magnetite in real industrial application.

2.   The structural characterization of CMNs is also necessary for comparison study with BMNs. How the Zeta potential of the samples was measured?

3.    The experimental data in the figures need error bar.

4.    The authors mentioned deconvolution for XPS, but I did not see the result in Figure3. The deconvolution should be added in Figure3.  

Author Response

Thank you for your comments.

Reviewer 3 Report

Kim and Roh have reported the Environmental Application of Biogenic Magnetite Nanoparticles to Remediate Chromium (III/VI)-Contaminated Water. Overall, I consider that a minor revision of this paper is required. There are key points to be corrected and improved.

My comments:

The novelty of the work is established. In the Introduction part, more recent references (2018-19) should be cited to make sure the study is well correlated with the recent data.

-Journal of Molecular Liquids, 2018, 258: 345-353.

- Critical Reviews in Environmental Science and Technology, 2016, 46.2: 93-118.

- International Nano Letters , 2014, 4.4: 129-135.

- Journal of Molecular Liquids , 2016, 213: 345-350.

- Studia Universitatis Babes-Bolyai, Chemistry , 2017, 62.2: 233-245.

- Physica E: Low-dimensional Systems and Nanostructures , 2019, 106: 150-155.

- Chemistry of Advanced Materials , 2016, 1.1.

- Journal of Molecular Liquids , 2016, 215: 221-228.

-Chemistry of Advanced Materials, 2016, 1.1.

- Potential applications of nanomaterials in wastewater treatment: nanoadsorbents performance. In: Advanced Treatment Techniques for Industrial Wastewater. IGI Global, 2019. p. 51-61.

- Nanomaterial Surface Modifications for Enhancement of the Pollutant Adsorption From Wastewater: Adsorption of Nanomaterials." In Nanotechnology Applications in Environmental Engineering, ed. Rabia Nazir, 143-170 (2019), accessed January 17, 2019. doi:10.4018/978-1-5225-5745-6.ch007

- Potential Applications of Nanomaterials in Wastewater Treatment: Nanoadsorbents Performance." In Advanced Treatment Techniques for Industrial Wastewater, ed. Athar Hussain and Sirajuddin Ahmed, 51-61 (2019), accessed January 17, 2019. doi:10.4018/978-1-5225-5754-8.ch004

Global Journal of Environmental Science and Management, 2015, 1.2: 149-158.

---------

Updated comments:

Comments to Author
Although the manuscript reports new findings for water treatments (Remediate Chromium (III/VI)-Contaminated Water) using very interesting adsorbent (Biogenic Magnetite Nanoparticles), the novelty of the work has not clearly stated in the introduction part. Following is the comment from my side to the author(s):
1- Needs to highlight the work novelty
2- Use more data analyses
3- Polish the manuscript English (try to improve language, check abstract attached I did correction)
4- Some references related to the aim of a manuscript for the comparison part should be added
*** use refs which suggested earlier

-----------

Author Response

Thank you for your comments.

Round 2

Reviewer 1 Report

I would suggest the authors to re-do the experiments to address the issue raised by my query 6.

Author Response

We sincerely appreciate your comments.

We think you have made a reasonable suggestion, but we are sorry to be unable to experiment again. So we have inserted a brief explanation in section 3.2  to help understand it:

 "The Cr(III/VI) concentrations were not measured between 48 hours and 2 weeks, but the data's absence did not have a significant effect on understanding the concentration decrease tendency since the major reaction of Cr removal occurred within 48 hours."

Thank you for your understanding.

Reviewer 2 Report

The authors have well responded the comments from the reviewers and the quality of this manuscript has been significantly improved. I recommend its publication now. 

Author Response

We sincerely appreciate your comments.